# Cervical-Vaginal Mucin in Fertility Assessment: CA125 as a Predictor of the Fertile Phase of the Normal Menstrual Cycle

**DOI:** 10.3390/medicina56060304

**Published:** 2020-06-20

**Authors:** A. Alexandre Trindade, Stephen J. Usala

**Affiliations:** 1Department of Mathematics and Statistics, Texas Tech University, 1108 Memorial Circle, Lubbock, TX 79409, USA; alex.trindade@ttu.edu; 2Department of Internal Medicine, Texas Tech University Health Sciences Center, 1400 S. Coulter Street, Amarillo, TX 79106, USA

**Keywords:** CA125, cervical mucus, natural family planning, pregnancy failure rate

## Abstract

*Background and objectives*: To evaluate the cervical-vaginal mucin, CA125, as a measure of fertility and possible method for natural family planning (NFP). *Materials and Methods*: Cervical-vaginal fluid (CVF) swab samples have been previously used to measure CA125, ‘Qvaginal CA125 levels’, as a function of time of cycle relative to Day 0, the first day of positive urine LH (luteinizing hormone). Data from 15 women, 20 cycles were used with an algorithm to establish the Fertile Start Day (FSD) for the cycles. The FSD was determined as either the second consecutive day of ≥20% Qvaginal CA125 rise or the first day of ≥400% rise. The interval, (FSD to Day +3), was used as the theoretical window of fertility, and conception rates assuming abstinence during this predicted period of fertility were computed using published day-specific probabilities of conception (*PoC*). *Results*: The mean FSD was Day −4.8 ± 0.5 (SE), 95% CI (−5.9, −3.7). The estimated pregnancy failure rate (PFR) with abstinence during [FSD, +3] was 10.7% ± 2.0% (SE), 95% CI (6.9%, 14.8%); with exclusion of one cycle with very low levels of Qvaginal CA125, the estimated PFR was 9.8% ± 1.9%, 95% CI (6.3%, 13.8%). Furthermore, the day-specific Qvaginal CA125 values were normalized to the respective peak Qvaginal CA125 for each cycle, and a mean normalized day-specific Qvaginal CA125 plot was generated. The first derivative of the mean normalized day-specific Qvaginal CA125 plot showed a significant increase between Day −4.5 and Day −3.5, which correlated with the mean FSD. *Conclusions*: A Qvaginal CA125-based method holds promise as a means to identify the start of the fertile window and may prove useful in NFP, especially when combined with available home hormonal fertility awareness kits.

## 1. Introduction

The majority of fertility awareness/natural family planning (NFP) methods encompass subjective observations of cervical mucus/cervical-vaginal fluid (CVF) [1,2,3]. The physical properties of cervical mucus/CVF that lend themselves to female self-observation can be even more accurate in assessing the fertile time than timing relative to the point of ovulation [4]. Given the importance of the physical cervical mucus/CVF paradigm in natural family planning methods, one can hypothesize that there are biochemical and molecular factors that could serve to objectify assessment of the fertile and infertile phases and prove useful to women practicing NFP.

CVF is composed of cervical mucus and vaginal fluid; vaginal fluid contains various cells, transudative solutes, and macromolecules [5,6]. The volume and elasticity of cervical mucus and CVF change during the menstrual cycle [7,8,9]. Large glycoproteins, mucins, are believed to render viscoelastic properties to cervical mucus and thereby CVF [10,11]. An abundance of candidate proteins and saccharide structures for fertility assessment have been revealed by proteome and transcriptome analysis of cervical mucus, cervical epithelium, and CVF [12,13]. MUC5B, MUC5AC, MUC6, MUC16, and MUC1, gel-forming and membrane bound [14,15,16], were identified using tandem mass spectra analysis [13,17].

MUC16—more commonly known as the tumor antigen, CA125—is a large glycoprotein classified as a membrane bound mucin, but it has been found in cervical mucus [16,18]. A cellular ‘sheddase’ activity has been proposed which releases the extracellular portion of CA125 [19]. Transcriptome analysis of the bovine estrus cycle indicated that CA125 was up-regulated during the estrogen phase [12]. The role of CA125 in cervical mucus/CVF and in the reproductive tract remains speculative, although it may serve in a ‘periciliary brush’ in cervical and vaginal epithelial needed to maintain its osmotic modulus and the integrity of the mucus layer [20].

A vaginal swab CA125 assay has been reported in which daily samples of CVF rendered a periodic pattern of CA125 that correlated with time of cycle: low CVF CA125 during the early preovulatory phase, a maximum generally during Day −4 to Day +1 (where Day 0 was defined as the first day of positive urine luteinizing hormone (LH)), and low levels during the luteal phase [21]. It was hypothesized that charting of such CVF CA125 levels, ‘Qvaginal CA125 levels’, could be used in a suitable algorithm to predict in real time the start of the fertile phase of the menstrual cycle and an interval of abstinence to avoid pregnancy. We tested this hypothesis with 20 normal menstrual cycles from 15 different women by an algorithm and predicted a Fertile Start Date (FSD) with a corresponding interval, (FSD to Day +3), of potential fertility and abstinence to avoid pregnancy. The probabilities of conception for sexual abstinence during the interval, (FSD, Day +3), for the 20 cycles, in 15 women were calculated using already established day-specific probability of conception (*PoC*) values relative to Day 0 [22,23]. The statistical analyses presented here indicate that methodologies based upon serial Qvaginal CA125 measurements hold promise as an adjunct for NFP.

## 2. Methods

### 2.1. Qvaginal CA125 Assay

Fifteen women, through a total of 20 ovulatory cycles, provided swab samples of CVF as previously described [21]. Characteristics of these subjects were previously published and are provided in the supplementary data [21]. The subjects began daily vaginal sampling with a cotton swab one to three days after the end of menses, which was Day −14 to Day −5 (−7.6 ± 0.6(SE)). After a vaginal swab sampling, the subjects would immediately immerse the cotton tip into 0.5 mL of deionized water and elute the CVF contents by a swirling motion for 3 seconds, then discard the swab after pressing the swab tip on the upper end of the microfuge tube to collect further absorbed fluid. At the end of a cycle, the microfuge tubes were briefly centrifuged to remove cellular debris, and the 0.5 mL sample volumes assayed for CA125 using the Siemens Immulite 2000 system as previously described [21]. The subjects kept a calendar record of collection including urine LH results, and the samples were indexed to Day 0, the day of first positive urine LH. The day-specific ‘Qvaginal CA125′ levels in U/mL corresponded to the concentration of the day-specific 0.5 mL sample [21]. Blood samples during the preovulatory, ovulatory, and luteal phases were obtained for serum estradiol, progesterone, LH, and FSH (follicle-stimulating hormone) measurements with the Siemens Immulite 2000 system.

### 2.2. Mean Normalized Day-Specific Qvaginal CA125 and Rate Of Change During Cycle

The day-specific Qvaginal CA125 values were normalized to their respective peak CA125 during the interval, (Start day of sampling, Day +3); the maximum CA125 levels during this interval ranged 4–5,740 U/mL. The mean normalized day-specific CA125 plot and its first derivative were computed using GraphPad Prism version 6 for Windows, GraphPad Software, San Diego, CA, USA, (www.graphpad.com).

### 2.3. Algorithm to Determine Fertile Start Day (FSD) and Calculation of Conception Rates

An algorithm was developed by inspection of the Qvaginal CA125 data to predict the FSD using the interval (−4, +1) as a marker for the published period of potential fertility [21]. The FSD was determined for each cycle either by the second consecutive day of increase in Qvaginal CA125 ≥20% or by the first day of increase in Qvaginal CA125 of at least 400%. The FSD was determined in 7 of 20 cycles by the second consecutive day of increase ≥20% and in 13 of 20 cycles by an increase of ≥4—fold. The FSD Qvaginal CA125 levels ranged 4–217 U/mL.

### 2.4. Statistical Methods

The pregnancy failure rate (PFR) was calculated assuming: ovulation occurred 24 hours after the urine LH signal and abstinence was maintained from the FSD until four days after the urine LH peak. On a scale of 0 to 1, we use the term probability of conception (*PoC*), so that PFR is equal to 100**PoC*.

Models for *PoC* are concisely summarized by Scarpa [24]. With regard to our present study, the pertinent ones are the so called day-specific *PoC* models with homogeneity (Scarpa, Section 5.1 [24]). Among these, the model of Schwartz et al. [25] is the most applicable given the nature of our data, and takes the form:(1)$$PoC=v\;[1-\prod_{k\in K\;}(1-\frac{p_{k}}{v}{)}^{x_{k}}]$$
where *K* is the set of days around ovulation when intercourse may result in conception. With the index *k* running over *K*, *x_k_* is a binary variable indicating whether intercourse occurred on day *k* (*x_k_* = 1) or not (*x_k_* = 0), while *p_k_* is the day-specific probability of conception (when intercourse occurs only on day *k* of the cycle). Finally, *v* is the cycle viability factor, which attempts to account for all other biological covariates (in addition to timing of the intercourse) that may affect conception. (A more sophisticated approach to this is to specifically model the dependence on all the measured covariates by introducing additional parameters.)

The homogeneity of this model refers to the fact that both *p_k_* and *v* (as well as the additional parameters relating to covariates, if any) are the same for all subjects in the study (homogeneous), whereas in reality this assumption may not be accurate. (Various modifications to account for different types of heterogeneity have been proposed; see Scarpa, Section 5.2 [24]). The values for *PoC* presented in the Appendix A were calculated from the estimated daily probability of conception (*p_k_*) values in Table 10 of Columbo and Masarotto, corresponding to the ‘European Centers’ column [23]. (This column contained the most comprehensive estimates, with K ranging from −8 to +2, based on a total of 5390 cycles.) We used the cycle viability value of *v* = 0.277, from Table 9 of Columbo and Masarotto corresponding to the basal body temperature (BBT) reference day [23]. The same value was used by Freundl et al. [22].

Our analysis utilized equation 1 to compute *PoC*, by setting *x_k_* = 0 for all days equal to FSD and beyond. An example will suffice to illustrate the nature of the calculations. For a subject with an FSD of −4, and since we assume abstinence is maintained from the FSD until four days after the urine LH peak, this means that only *p_k_* values corresponding to *k* = {−8, −7, −6, −5} will be used to obtain:*PoC* = *v*(1 − (1 − 0.003/*v*) × (1 − 0.14/*v*) × (1 − 0.027/*v*) × (1 − 0.068/*v*)) = 0.100(2)
where *v* = 0.277 was utilized.

## 3. Results

### 3.1. The Rate of Change Signature for Qvaginal CA125

In order to better understand and quantitate the relation of Qvaginal CA125 to time of cycle, the day-specific Qvaginal CA125 levels were normalized to the peak level in the interval, Day −14 to Day +3. The mean peak day for this interval was Day −0.8 ± 0.5(SE), range −7 to +3, and the mean peak Qvaginal CA125 level was 630.0 ± 295.5(SE) U/mL, range 4.9–5,740 U/mL. However, even with this considerable variance, the normalized plot (Figure 1, top) showed a significant rate of change as demonstrated by its first derivative analysis (Figure 1, bottom). Both the rise and rate of rise in slope increased sharply between Day −4.5 and Day −3.5, in close agreement with the mean FSD of Day −4.8 (below). The maximal rate of change for the 20 cycles, 15 patients, occurred at Day −3.5 and Day −2.5; there was a negative inflection point between Days −1.5 and −0.5. These results suggest the rate of change in Qvaginal CA125 could serve as a metric for the fertile window before ovulation.

### 3.2. Fertile Start Day (FSD) for 20 Cycles, 15 Patients, and Estimation of Pregnancy Failure Rates with Qvaginal CA125

The FSD using the algorithm (Methods) are shown in Figure 2a. The mean FSD was Day −4.8 ± 0.5(SE) (95% CI: −5.9, −3.7). To establish a fertile interval, the FSD data were used with the fact that the *PoC* is quite low three days after positive urine LH (Day +3). Therefore, the PFRs for abstinence during FSD to Day +3, assuming daily sexual intercourse prior to FSD, were computed as a function of FSD for the cycles.

The mean PFR was 10.7% for *n* = 20 cycles, in 15 different patients (Table 1). Of note, one patient had very low Qvaginal CA125 levels, barely above background; this cycle with the present algorithm rendered an FSD of Day −2. With exclusion of this outlier, the meaAn PFR was 9.8% (Table 1). In the quantification of the uncertainty for the mean PFR, we opted for a bootstrap 95% confidence interval (nonparametric BCa), which places it between 6.9% and 14.8% (*n =* 20) or 6.3% and 13.8% (*n =* 19). Nonparametric bootstrap inferential procedures make few assumptions about the data (in this case, only that the cycles are independent) and constitute the methodology of choice in serious scientific research due to their versatility and robustness to underlying assumptions [27].

Two estimates of the probability density function of *PoC* are displayed in Figure 2b. The shaded grey region is a histogram, while the solid line is a kernel density smoothed estimate [27]. The salient feature in these is the bimodality of the distribution. There appear to be two distinct regimes: low with *PoC* in the range (0–0.10), and high with *PoC* in the range (0.20– 0.30). In essence, the reason for this abrupt separation is the discreteness inherent in equation 1. In Figure 2c, which plots the *p_k_* estimates we fed into the model, we see that there is a sharp exponential increase in daily *PoC* at FSD in the range (–5, –3). This discreteness then leads to the consequently large jump in *PoC* in the same region (Figure 2d), thus causing the bimodality in Figure 2b. Figure 2d displays the relationship between FSD and *PoC* over the entire range of FSD values, for which there is a positive estimate of *p_k_* in Table 10 of Colombo and Masarotto [23]. The *PoC* starts at 0 at FSD of −8 and eventually plateaus off at a maximum of *v* = 0.277 by FSD of +2. 

### 3.3. Preovulatory Estradiol Levels and Predicted Infertility Intervals

Serum estradiol was measured during the preovulatory and periovulatory phases, one to three samples per cycle, and, by Day −5, serum estradiol was greater than 183.5 pmol/L (50 pg/mL); serum estradiol was less than 183.5 pmol/L (50 pg/mL) on Day −8 and earlier (Figure 3).

We hypothesized that the FSD computation could reduce the length of abstinence in NFP. The longer cycles with longer preovulatory phases had earlier sampling start days. The preovulatory infertile interval was considered the sampling start day to the FSD; that is, abstinence would begin on the FSD. The interval, (Sampling start day, FSD), was plotted as a function of sampling start day (Figure 4). In general, the FSD signal did theoretically provide reduced abstinence for NFP.

## 4. Discussion

This is the first report of a cervical mucus and vaginal fluid macromolecule that has been considered for a practical method for fertility awareness and natural family planning. The theoretical probabilities of pregnancy—that is, the pregnancy failure rates with the Qvaginal CA125 method—were calculated using day-specific pregnancy probabilities indexed to the first day of positive urine LH, Day 0. It should be noted that the theoretical pregnancy failure rate, 10.7% ± 2.0%(SE), because of the index based on the urine LH signal, may have a high bias, since there are clinical trials which indicate that cervical mucus/CVF physical parameters can be a better predictor of the fertile window than timing with hormonal values [4,28,29]. Assuming that CA125 (the MUC16 mucin) tracks fertile cervical mucus, a day of low Qvaginal CA125 and low rate of change even in the fertile window of Day −4 to Day +1, would putatively indicate less fertility than that solely considered by reference to urine LH signal.

The use of Qvaginal CA125, and perhaps other CVF macromolecules, for personal fertility monitoring is not envisioned as a stand-alone technology. The hormonal assays, urine LH or urine pregnanediol-3-glucuronide (PdG), are powerful signals for the ovulatory and post-ovulatory phases of the menstrual cycle and home assays are already available [1,22,30,31]. The major problem with the effectiveness and acceptability of natural family planning methods presently appears to be in identifying a start day of fertility (i.e., the FSD) in the follicular/preovulatory phase of the cycle in order to provide both a low conception rate and a minimum period of abstinence. In this regard, in terms of the effectiveness of the personal fertility monitors for pregnancy avoidance, the Marquette Method is cited to be 99.4% for correct use, 89.4% for typical use, and the Persona 89%–94% [1,22]. The estimated effectiveness of the Qvaginal CA125 method, 89.3–90.2%, before further optimization, is in this range. A future Qvaginal CA125 test strip could potentially be incorporated into a ‘double-check’ form of NFP, which could include an apparatus, such as the Persona and urine PdG strips.

Three important issues are left for optimization of the Qvaginal CA125 assay for development as a personal fertility monitor: strengthening the algorithm for prediction of the fertile window, a home test strip (lateral flow assay), and normalization of Qvaginal CA125 levels to account for variable collection. Test strips for CA125 are already listed on the internet as available from China (Aluxbio Co. and Quicking Biotech Co., Shanghai, China). To reduce sample variation, normalization of the Qvaginal CA125 result from CVF sampling to a house keeping gene product or total protein could be accomplished. This would have to be incorporated into any test strip and measurement apparatus. In addition, the algorithm used for the present study was not uniquely selected, and it is believed that more powerful algorithms can be derived to increase the sensitivity and specificity of the Qvaginal CA125 metric to reduce both the rate of pregnancy and level of abstinence.

## 5. Conclusions

The analysis herein involved normal ovulatory menstrual cycles of length days 21–35 and indicated a Qvaginal CA125-based method holds promise as a means of fertility assessment. It remains to be determined how well the Qvaginal CA125 method can be used in irregular cycles, during breast feeding, and in the context of other conditions which could potentially modify CA125 secretion, such as endometriosis and hepatic changes.

## Figures and Tables

**Figure 1 medicina-56-00304-f001:**
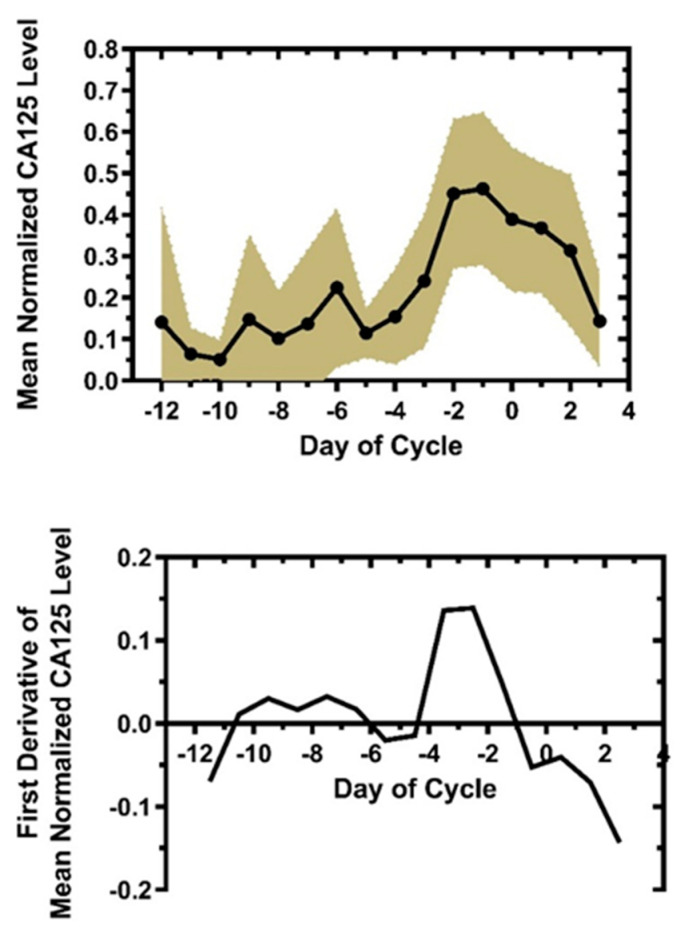
Mean normalized day-specific Qvaginal CA125 levels (top, shaded 95% CI) with the first derivative of this plot (bottom). Results are from 20 cycles, in 15 different subjects. Qvaginal CA125 levels were normalized to the respective maximum level in each cycle during the interval, (Start of sampling, Day +3), and the mean day-specific normalized Qvaginal CA125 curve generated. The rate of change of the mean normalized Qvaginal level, the first derivative, was then determined. The major rise occurred between Day −4.5 and Day −3.5; first derivatives were −0.014 and +0.136, respectively. Day 0 is first day of positive urine luteinizing hormone (LH).

**Figure 2 medicina-56-00304-f002:**
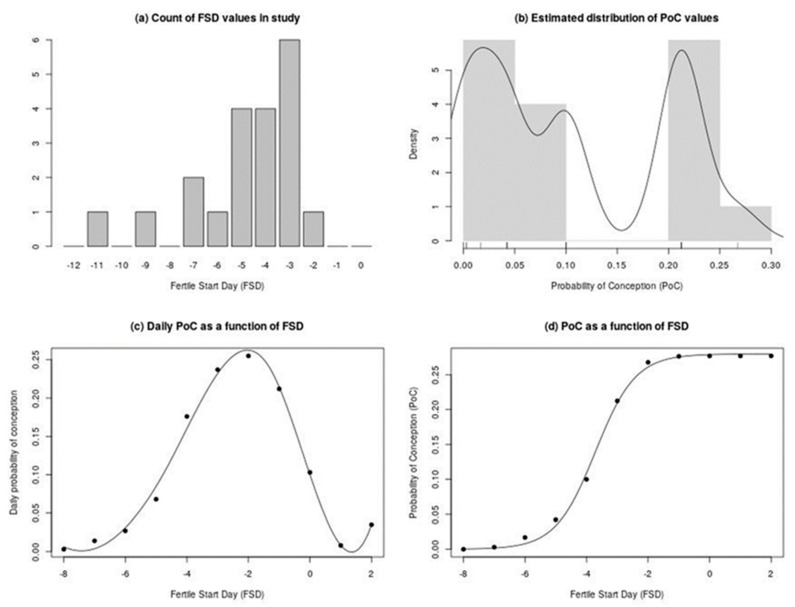
(**a**) Distribution of Fertile Start Day (FSD) for the 20 cycles, in 15 subjects, during the preovulatory phase. The mean FSD was Day −4.8 ± 0.5(SE) with 95% CI of (−5.9, −3.7). The cycle with FSD, Day −2, was an outlier in terms of low Qvaginal CA125 levels. Day 0 is first day of positive urine LH. (**b**) Two estimates of the distribution of probability of conception (*PoC*). The shaded grey region is a histogram. The solid line is a kernel density smoothed estimate [26]. (**c**) Daily *PoC* estimates from the European Centers column in Table 10 of Colombo and Masarotto [23]. The solid line is a loess smoothed rendering of the points [27]. (**d**) *PoC* as a function of FSD, calculated according to our algorithm, whereby abstinence is maintained from the FSD onward. The solid line is a regression curve smoothed rendering of the points by fitting a logistic growth curve [27].

**Figure 3 medicina-56-00304-f003:**
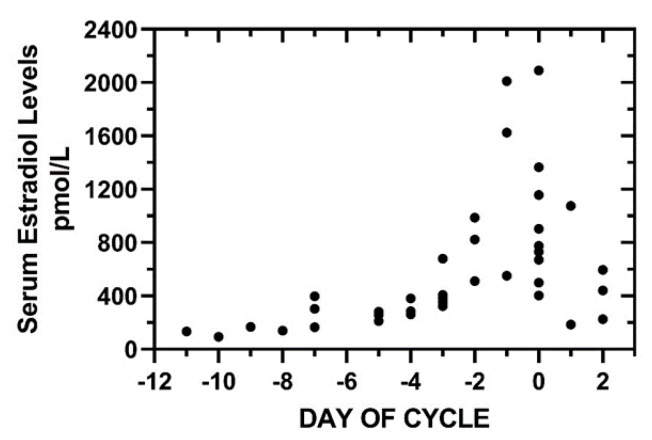
Serum estradiol levels during the preovulatory interval to Day +2. One to three estradiol levels per cycle were obtained in the 20 cycle cohort. The serum estradiol levels (pmol/L) were determined using the Siemens Immulite 2000 system. Estradiol levels during the interval, (−11, −8), were <183.5 pmol/L (50 pg/mL). By Day −7, estradiol levels had reached 165.2–396.4 pmol/L (45–108 pg/mL). Day 0 is first day of positive urine LH.

**Figure 4 medicina-56-00304-f004:**
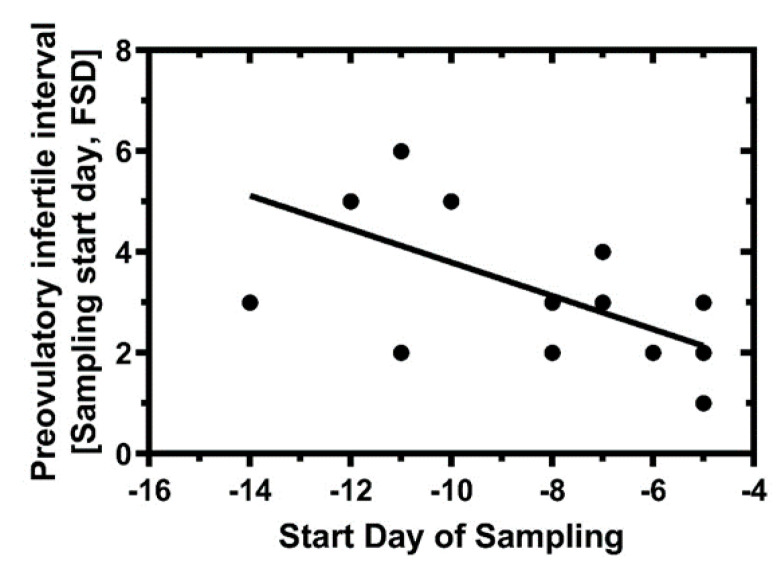
The Fertile Start Day (FSD) algorithm generally provides a greater interval of predicted preovulatory infertility for longer cycles. The ‘safe period’ to avoid pregnancy is predicted to be the interval beginning with the start day of sampling and ending with the FSD, (Sampling start day, FSD); that is, the period of predicted fertility begins on the FSD. A linear regression of this interval length as a function of sampling start day showed a significantly negative slope with an R-squared = 0.4291, *p* = 0.0017. Day 0 is first day of positive urine LH.

**Table 1 medicina-56-00304-t001:** Descriptive statistics for probability of conception (*PoC*) with CA125 method.

	Fertile Start Day (FSD)	*PoC* (*n* = 20 Cycles)	*PoC* (*n* = 19 Cycles, No Outlier)
Number of Cycles	20	20	19
Minimum	−11.0	-	-
Maximum	−2.0	-	-
Mean	−4.8	0.107	0.098
Standard Error of Mean	0.5	0.020	0.019
Lower 95% CI of mean	−5.9	0.069	0.063
Upper 95% CI of mean	−3.7	0.148	0.138

Fertile Start Day (FSD) is the second consecutive day of rise in Qvaginal CA125 by 20% or the first day of rise by 400%. Day 0 is first day of positive urine LH. *PoC* was calculated for abstinence during interval, [FSD to Day +3], using day-specific probabilities of conception from Columbo and Masarotto [23].

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
