# Peer review of "Cervical-Vaginal Mucin in Fertility Assessment: CA125 as a Predictor of the Fertile Phase of the Normal Menstrual Cycle"

_medicina, 2020, doi:10.3390/medicina56060304_

Round 1
Reviewer 1 Report
Dear Authors and Editor,
thank you very much for giving me the possibility to review the article “Cervical-vaginal mucin in fertility assessment: MUC16/CA125 as a predictor of the fertile phase of the normal menstrual cycle” by Trindade et al. This is a very technical and methodological paper regarding the feasibility to use the MUC16/CA125 levels to detect the fertility period of a woman.
As first instance I can note that terminology MUC16 compares only in the title and not in the abstract, so I suggest to be consistent. Generally the authors should avoid the use of so many abbreviations, since the readers can have few difficulty in the flowing of the text. Introduction should be better organized, for ecxample the aims of the study should be at the end, and results should not be reported. Further, half od the introduction can be moved to Methods and one third to Results.
The first criticism is that the swab was taken by the subjects and hence it could be biased in the collection. The second crucial criticism is that the study did not report the characteristichs of the population even though they have declared that they are homogeneous (as assumed by v covariates). Notably, author should state if they considered also the partners, if the women had previous viable pregnancy, regardless their homogeneity and thus they should clear discuss the biological covariates considered. For example, does this test work better in young women? The age range of the patients is 21-35 and this is a great variation. Further, maybe the extra 5 cycles derived from the youngest patient or from the older, of which we do not know the clinical history.
Authors should be more consistent with this definition “The FSD was determined for each cycle either by the second consecutive day of increase in Qvaginal CA125 ≥20% or by the first day of increase in Qvaginal CA125 of at least 400%”, is this arbitrary or not? Further, in my opinion the authors can be less technical to enhance the readability also by pure clinicians not used to manage so many rates, and formulae and not by statistics.
The results are far away more rich that what stated in the methods, and sometimes the results give rise to further speculation “We hypothesized that the FSD computation could reduce the length of abstinence in NFP.” So I suggest to better organize the part of the results, or to shrink it.
In the discussion the authors should discuss about future studies, what is needed to go ahead with the project, what are the strength and the limits. For example, what about patients with endometriosis? Or who had surgery with endometriosis? The paper is very technical and I really appreciate the methods, but the authors should face with patients from the real world. In this study using “homogeneous” patients, the effectiveness of Qvaginal CA125 is inferior to Persona or Marquette Method. Authors should discuss how they think to overcome the limitation when this test would be used outside research, and what they mean for optimization.
I recommend to simplify the article if the authors and the editor aim to reach a wide diffusion among gynaecologists and to clearly state the limits.
Best regards
Reviewer 2 Report
I think it is an interesting, original and well-written paper but I am not able to judge the mathematical models that were employed.
Here are my comments:
The finding of a biochemical marker that can be easily determined to predict the fertile window is relevant and original. In fact, the developmnent of a simple method of self assessment of the fertility window might reduce the subjectivity and increase the accuracy of NFP, empowering women who have contraindications or do not want to use medical contraception methods.
The paper is well written and easy to read. However I am not able to evaluate the mathematical model used.
The discusion is clear and empahzises the main drawbacks of the study suggesting how they can be overcome with further research.
Briefly, provided taht the mathematicla model can be validate by another reviewer.
Round 2
Reviewer 1 Report
Thank you very much for the opportunity to review again your paper. I went through the responses of the authors and I found that not my suggestions have been considered partially. However, I think methodology and aims of the paper are clear, even though I suggest the author to give more prospective regarding the future devolopment of this method especially in the context of a real utility. Again, how can method can be useful as complementary? Please explain in the text and give clinical hypotesis of utilization. What about patients with endometriosis or undiagnosed ovarian cancer or hepatic problems?
Again I found the introduction very long, and I ca see that few sentences should be properly moved to M&M and few sentences of M&M can be moved to Results.
The complex technical writing style and content of the article are not prone to be kindly used from physician, but mostly by technicians and statisticians, so, as the authors themself declared, it's up to the editorial board decide if this article, its methodology and content are adequated for the journal.
Kind regards.
